# Multimodal Clustering of Students Using Social Media Profiles to Analyze Academic Performance

**Sergei S. Gorshkov**
Department of Computer Science
National Research University Higher School of Economics
Moscow, Russia
sgorshkov@hse.ru

## Abstract

This study explores the relationship between students' digital behavior and academic performance using a multimodal clustering framework applied to social media profiles. Unlike traditional self-reported surveys, it objectively analyzes students' social media footprints by integrating text and image embeddings through state-of-the-art machine learning models, including Sentence-BERT, CLIP, BERTopic, HDBSCAN, and KMeans++ clustering. Using a dataset of 2,909 students, the study identifies 52 distinct behavioral clusters, revealing that engagement in educational and scientific content is associated with higher academic performance, while entertainment-focused digital habits are associated with lower GPAs. Statistical analyses confirm significant differences across clusters, highlighting structured associations between digital behavior patterns and academic outcomes. These findings contribute to AI-driven education analytics and demonstrate how multimodal machine learning and big data analytics can support large-scale analysis of student digital behavior in relation to academic performance. Future research should explore longitudinal trends to refine the interpretation of these patterns and further extend intelligent learning analytics.

## 1 Introduction

The rapid digitalization of society has transformed how individuals interact, learn, and engage with online content. Artificial intelligence (AI), machine learning (ML), and big data analytics are increasingly used in education for adaptive learning, recommendation systems, and academic analytics. Social media platforms shape how students consume information, express opinions, and engage with communities, which makes them a potentially informative source for studying relationships between online behavior and academic outcomes Cui et al. (2019), Shahiri et al. (2015). However, studying this relationship is challenging, as digital engagement is complex and multifaceted Junco (2012). Traditional approaches often rely on self-reported surveys, which may be biased and limited in scope Salinas-Chipana et al. (2024), Alyahyan & Düştegör (2020). Moreover, existing research typically examines either textual or visual content in isolation, which may obscure richer behavioral structure.

This study addresses these challenges through a multimodal analytical approach that combines natural language processing, computer vision, clustering, and statistical analysis to examine how social media behavior relates to academic performance. The paper develops and evaluates an integrated pipeline for large-scale educational behavior analysis on real-world multimodal data, with the goal of identifying stable student groupings and their differences in academic outcomes. The primary objectives of this research are:

- To develop a stable and interpretable clustering model based on students' social media activity.

- To examine the relationship between these clusters and academic performance (GPA).

- To identify key textual and visual engagement patterns that characterize different student profiles.

To achieve these objectives, we employ state-of-the-art machine learning techniques, including Sentence-BERT for text embeddings, CLIP for image representation, and BERTopic for topic modeling. For clustering, we utilize a combination of HDBSCAN for image categorization, K-Means++ for final student grouping, and UMAP for dimensionality reduction in order to balance computational efficiency and interpretability.

More broadly, this study contributes to the AI-in-education landscape by demonstrating how multimodal analytics can be used to examine student digital behavior in relation to academic outcomes. The proposed framework may support exploratory educational analytics, including the analysis of engagement patterns, the identification of heterogeneous student profiles, and the development of data-informed hypotheses for future learning support tools Jimenez Martinez et al. (2024). In this way, multimodal behavioral clustering can serve as a scalable analytical approach for studying student differences in large digital environments.

## 2 LITERATURE REVIEW

The integration of digital technologies in education has significantly impacted student engagement and academic performance Ahmad et al. (2024). Research highlights both positive and negative effects, where structured digital engagement enhances learning outcomes, while excessive digital distractions hinder academic success.

Several studies have investigated these effects. In Giunchiglia et al. (2018) a negative correlation was found between social media usage and GPA, particularly when engagement involved non-educational content. Similarly, Pérez-Juárez et al. (2023) reported that multitasking with digital devices in learning environments reduces student focus and academic performance. However, structured digital literacy programs have been linked to improved learning outcomes, as a meta-analysis found a strong association between digital competencies and academic success Li et al. (2025).

A systematic review of machine learning models for academic performance prediction found that deep learning and ensemble methods effectively identify at-risk students based on online engagement patterns Akpen et al. (2024). Digital game-based learning has also shown promise, where students engaging with educational gaming environments demonstrated higher motivation and problem-solving skills Li et al. (2024).

Despite the growing body of research on digital learning, there remains a notable gap in studies directly linking students' social media profiles to academic performance. Existing research has explored the predictive potential of social media text data for academic success, demonstrating that linguistic patterns in students' posts could serve as predictors Smirnov (2020). However, comprehensive investigations that systematically analyze how students' social media behaviors and digital personas correlate with their educational achievements remain scarce.

Moreover, recent systematic reviews provide an up-to-date examination of artificial intelligence in higher education Wang et al. (2024), Crompton & Burke (2023), including intelligent tutoring systems, automated assessment, and personalized learning, highlighting both the benefits and challenges associated with implementing AI-driven technologies in educational settings.

## 3 PROPOSED METHOD AND RESULTS

To analyze the relationship between students' digital behavior and academic performance, we developed a multimodal analytical approach that integrates text and image embeddings with clustering and statistical evaluation. The method consists of four main stages: feature extraction from textual and visual data, vector profile creation, student clustering based on multimodal embeddings, and statistical analysis of GPA differences across clusters.

## 3.1 Data Collection and Processing

The study was conducted using publicly available profile data from 3,540 students at Tomsk State University. Informed consent was obtained, ensuring that the data was anonymized and presented in aggregate form. Students voluntarily provided their VK social network profile identifiers, which served as the key source of multimodal data. The data collection process was structured to ensure both the accuracy of extracted information and compliance with ethical and privacy standards. After retrieval, all profile identifiers were anonymized. Ethical approval was granted by TSU's Faculty of Psychology Ethics Committee in September 2020 for a five-year period, requiring no further review within this timeframe.

The dataset was constructed using three primary sources: textual data, visual data, and academic metadata. Textual data comprised personal posts and subscriptions to online communities, with an average of nine unique posts collected per profile, along with descriptions and recent content from the subscribed communities. To preserve relevance while reducing redundancy, only the latest 20 non-advertising posts from each community were retained. Visual data included profile pictures and uploaded images, averaging 14 images per student. Only images from the last five years were considered to maintain temporal relevance. Additionally, visual content from subscribed communities was analyzed by extracting the 20 most recent images per group. The metadata component included students' GPA, faculty, and level of study. To reduce redundancy, only the 20 most relevant communities per user were retained, with community-based embeddings computed separately before being integrated into individual student profiles. This sampling strategy was pragmatic, but it does not enforce identical temporal windows across communities with different activity levels.

Data was stored in a MongoDB system, with structured extraction via the VK API and preprocessing handled by Python libraries. Textual data underwent cleaning procedures, including duplicate removal and exclusion of low-information content. Image data was retrieved asynchronously and enriched with metadata such as resolution, and upload date. Profiles with fewer than three posts or images were excluded, reducing the sample to 2,909 students. A comparative analysis showed no significant differences in grade distribution between excluded and retained students. We also explored whether social media activity, measured by content volume, correlated with academic performance. Students were grouped by content levels: fewer than 10 posts/images, 10–20, 20–40, and more than 40. No statistically significant differences were found, indicating that VK activity volume alone does not explain academic performance differences in this dataset.

## 3.2 Feature Extraction from Digital Data

To transform students' online activity into a structured numerical format, both textual and visual content were embedded into vector representations. For each student, we compiled a list of posts and photos from their personal page, along with posts and images from popular communities they were subscribed to. Initially, embeddings were computed for each individual text and image. We then conducted a series of experiments to determine the most effective method for combining these embeddings. Two primary approaches were considered:

- Category-Based Averaging. Mean embeddings were computed separately for personal posts, personal images, community posts, and community images, then concatenated. While this preserved category distinctions, it led to high-dimensional, computationally costly representations.

- Content-Type Averaging. Mean embeddings were computed for all images and all texts separately, resulting in a simpler, more efficient representation.

The second approach demonstrated better performance. A likely reason is the complementary nature of personal page content and community content. Typically, personal pages often contain personal experiences, travel photos, educational materials, and reflections, while community content reflects broader interests such as sports, gaming, and celebrities. Many personal posts are reposts from communities, further blurring distinctions. Using separate vectors for each category led to redundant representations, reducing model reliability. Given limited data, high-dimensional feature spaces risked introducing spurious correlations. Thus, we adopted a single embedding for all images and another for all texts.

Textual features were extracted using Sentence-BERT (all-mpnet-base-v2) Reimers & Gurevych (2019), which generates 768-dimensional embeddings. Each student's textual representation was formed by aggregating embeddings from posts, subscriptions, and community descriptions. Post embeddings were computed individually and averaged across all posts, while community descriptions were processed similarly, with embeddings aggregated across subscriptions.

Visual features were extracted using CLIP (ViT-B/32) Radford et al. (2021), a model designed to align images and text in a shared embedding space. Each image was converted into a 512-dimensional embedding, forming a unified vector representation for each student's uploaded content. Separate embeddings were maintained for personal profile images and community-based visuals to distinguish individual preferences from broader visual trends.

## 3.3 TOPIC MODELING

As in the previous section, we examined two approaches – one that separately processed content from personal pages and communities, and another that merged all data into a unified representation. The latter approach once again demonstrated superior performance. Additionally, after interpreting the resulting topics and clusters, we found that the overall quality was higher when all textual data was treated as a single source. This approach led to fewer uninterpretable and redundant topics.

To analyze students' textual activity, BERTopic Grootendorst (2022) clustered Sentence-BERT embeddings into 71 topics, excluding an outlier. The topic count was algorithmically determined and validated via keyword analysis. After manual review, six topics were merged based on lexical similarities, resulting in a 65-dimensional probability distribution per student.

The largest topic clusters included categories that we labeled, for brevity, as "memes", "music", "education", "family and relationships", "movies", "politics", "small businesses in the city", "photo hashtags", "literature", and "nature". It is worth noting that some thematic areas, such as sports, were split into several smaller topics, which, if combined, would rank among the top five. Similarly, themes such as "concerts" and "city events" were treated as separate topics, even though they could have been merged with broader categories. This approach allowed us to capture nuanced variations within students' interests while maintaining an interpretable structure.

## 3.4 IMAGE CLUSTERING

For visual data, HDBSCAN Campello et al. (2013) was used to cluster image embeddings without requiring a predefined number of clusters. However, careful hyperparameter tuning was necessary to avoid excessive fragmentation or merging of dissimilar samples. We tested minimum cluster sizes from 10 to 100 and minimum samples per core point from 5 to 50, selecting the best parameters based on the Silhouette Score. The optimal configuration was a minimum cluster size of 25 and a minimum sample count of 5, which was further validated using the Davies-Bouldin Index (DBI). Since HDBSCAN lacks explicit centroids (unlike K-Means), cluster interpretation followed a structured approach:

- Selecting core images with the highest confidence scores for belonging to a specific cluster.
- Identifying the 10 closest images via cosine similarity to assess visual coherence.
- Performing manual analysis to flag clusters with highly diverse or unrelated images.

This process identified 62 clusters, with 14 classified as noise clusters, comprising 17% of images – an optimal balance between data retention and overfitting prevention. The largest non-noise image clusters encompassed themes such as "human portraits", "nature and landscapes", "group photos", (including family gatherings, friend groups, and social events) "memes", "pets", "images with short text" (primarily event announcements, quotes, and informational infographics), "food", "gaming and anime", "miscellaneous images" (a broad category covering digital artwork, car photography, and random collages), and "sports events".

For many of these broad themes, subtopics were identified within smaller clusters. Also, several significant topics, such as "politics" and "news", did not form distinct clusters under any parameter settings. This suggests that political content is difficult to distinguish purely based on image embed-

dings without supervision. However, it is reassuring that topic modeling successfully captured these themes within textual data, compensating for their absence in image clustering.

## 3.5 STUDENT CLUSTERING

Given the high dimensionality of the constructed feature vector (768 dimensions for text embeddings, 65 for topic distributions, 512 for image embeddings, and 48 for image category probabilities), direct clustering in this space is computationally inefficient. To address this, Uniform Manifold Approximation and Projection (UMAP) McInnes et al. (2018) is applied to embeddings to reduce dimensionality while preserving the semantic structure of the data. Various target dimensions were tested, ranging from 10 to 200, and evaluated based on variance retention and clustering performance using K-Means++ Arthur & Vassilvitskii (2007) with Silhouette Score calculations. Lower dimensions resulted in excessive information loss, whereas higher dimensions led to overfitting and an unnecessarily complex data structure Rahamat Basha & Rani (2019). An optimal balance was identified at 40 dimensions.

UMAP was chosen over PCA and t-SNE due to its superior ability to retain both global and local structures, ensuring efficient processing while maintaining meaningful relationships among students. After dimensionality reduction, each student is represented as a unified feature vector integrating multimodal information through the concatenation of:

- Sentence-BERT textual embeddings, reduced via UMAP to 40 dimensions, capturing the semantic features of a student's posts and subscribed communities.

- BERTopic probabilistic topic distribution with 65 dimensions, indicating the likelihood of a student engaging in different thematic clusters.

- CLIP image embeddings, reduced via UMAP to 40 dimensions, encoding visual information from profile pictures and community media.

- HDBSCAN probabilistic image category distribution with 48 dimensions, representing the likelihood of a student's images belonging to different image-based thematic clusters.

This multimodal representation provides a comprehensive digital fingerprint, integrating textual interests, visual preferences, and behavioral patterns. With dimensionality reduction applied, clustering is performed using K-Means. The number of clusters $k$ is determined through multiple validation methods. The Elbow Method identifies an inflection point around $k \approx 50 - 55$, while Silhouette Score optimization finalizes $k = 52$, ensuring distinct and interpretable student segments.

The distribution of students across clusters is presented in Figure 1.

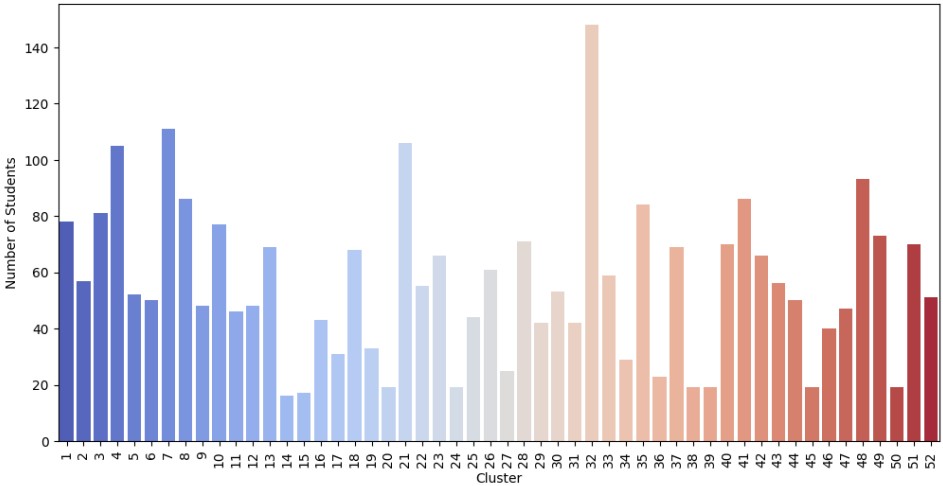

Figure 1: Number of students per cluster.

To validate the reliability of the clustering process, several techniques are employed. Centroid-based visual inspection involves extracting the ten students closest to each cluster centroid to assess behavioral consistency. A cluster size histogram shows that the two smallest clusters contain fewer than 20 students but still exhibit meaningful patterns. No cluster dominates the dataset, ensuring balance. Thematic distribution analysis with BERTopic and HDBSCAN confirms significant differences across clusters, reinforcing thematic validity.

K-Means is preferred over HDBSCAN and Agglomerative Clustering, as these methods aggressively remove "noisy" points – useful for filtering rare or anomalous images but unsuitable for student clustering, where every student must belong to a group. Alternative methods struggled with the heterogeneous feature space, while K-Means handled it effectively.

Cluster compositions are interpreted through a hybrid approach:

- Extracting the top-3 topic probabilities from BERTopic and top-3 image category probabilities from HDBSCAN for each cluster.
- Analyzing the average topic distribution across the cluster if topics were too spread out.
- Identifying dominant topics when over 90% of students in a cluster had a high probability ($> 0.9$) for the same themes.
- Verifying consistency by inspecting the top-10 students closest to the centroid.

Approximately 40% of clusters were highly coherent, while the remaining clusters were more mixed. Among the cohesive clusters, more than half had consistent topics across both text and image embeddings at similarly high or low levels. While it might seem beneficial to merge similar topics from text and images, in practice, content differences between the two modalities were significant, making merging infeasible.

### 3.6 ACADEMIC PERFORMANCE ANALYSIS ACROSS CLUSTERS

After clustering 2,909 students into 52 groups, we examined the distribution of GPA across clusters to determine performance differences. The GPA distribution across clusters is shown in Fig 2.

The distribution of students across clusters is presented in Figure 1.

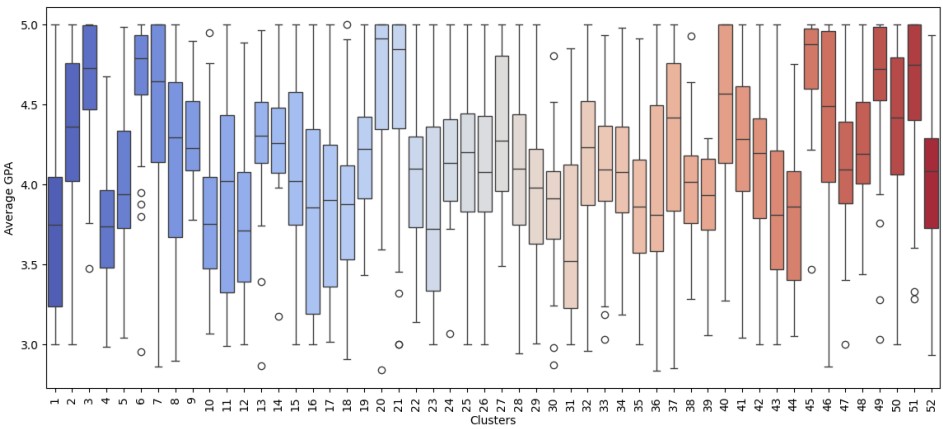

Figure 2: GPA distribution across clusters.

Following the methodology described above, we conducted an in-depth analysis of the obtained clusters, focusing on the interpretation of those that contained the highest-performing and lowest-performing students. Additionally, we examined clusters with the widest GPA range. Some clusters, particularly the largest one, could not be meaningfully interpreted, as they lacked dominant themes or discernible patterns. Below, we provide descriptions of selected clusters that were successfully interpreted, where the dominant themes were clearly distinguishable. Clusters with High-Achieving Students:

- Foreign Language Learning. This cluster is strongly influenced by topics related to foreign language vocabulary and linguistic studies, as identified through topic modeling. The images predominantly contain text or neutral content, with no dominant image clusters.

- Natural Sciences. The text content in this cluster is focused on subjects related to STEM fields, biology and chemistry. The associated images primarily contain text or depict various inanimate objects on plain backgrounds.

- Education and University Engagement. This cluster reflects strong student involvement in academic life and a broad spectrum of intellectual interests. The dominant image types include group photos (often depicting student gatherings or study sessions) and event posters.

Clusters with Low-Achieving Students:

- Memes and Informal Humor. The textual content in this cluster revolves around humor, memes, and slang, including explicit language. The images are similarly unstructured, dominated by memes and random visuals without a clear thematic focus.

- Gaming. The texts in this cluster frequently contain terminology related to video games. The images are mostly digitally illustrated characters, often from animated or gaming contexts.

- Celebrity Culture, Fashion, and Glamour. This cluster is characterized by content focused on celebrities, fashion trends, and luxury lifestyles. The dominant image type consists of professionally shot portraits of individuals, typically non-selfies. These images often feature celebrities or models promoting products.

Clusters with the Broadest GPA Distribution:

- Music and Concerts. The textual content in this cluster relates to music and musicians. No dominant image category was identified, as music-related posts often include stock photos or album covers, making it difficult to classify them into a specific visual category.

- Animal-Related Content. The text-based themes in this cluster are centered around animals, while the images predominantly feature various animal-related photographs.

- Volunteering and Community Events. The text content is closely linked to volunteering and event organization. The dominant image category includes group photos, typically taken at social events and gatherings.

Overall, text-based topic modeling produced clearer cluster distinctions associated with GPA differences than image-based clustering. This discrepancy may stem from two main factors: (1) the image embedding model used may not have been sufficiently sensitive for nuanced interpretation, and (2) many social media images are difficult to categorize meaningfully. Users often attach visually appealing images that may not directly relate to the accompanying text or simply upload images containing text summaries of the post itself. Notably, broad themes such as nature, travel, family, and relationships did not strongly dominate any specific cluster, suggesting that these interests are widespread among students and are not strong distinguishing factors in this type of digital behavior analysis.

### 3.7 STATISTICAL ANALYSIS OF GPA DIFFERENCES ACROSS CLUSTERS

To assess GPA differences across clusters, we applied Analysis of Variance (ANOVA), which confirmed strong GPA disparities across clusters ($p < 10^{-100}$). Since ANOVA only identifies the presence of differences without specifying their location, we conducted post-hoc pairwise comparisons. To validate these comparisons, we tested for normality using the Shapiro-Wilk test and visual inspections (histograms, Q-Q plots). The results ($p < 0.05$ in most clusters) suggested deviations from normality, particularly in some clusters. Given this, we complemented Welch's t-tests with the Mann-Whitney U-test to ensure robustness against non-normality.

A key result was that all direct comparisons between high- and low-achieving clusters showed statistically significant GPA differences ($p < 0.001$ in each case, adjusted for multiple comparisons). Students in clusters focused on sciences, foreign languages, and university engagement consistently outperformed those in clusters centered on memes, gaming, and celebrity culture. Beyond these

direct comparisons, we identified over 25 clusters with significant GPA differences across multiple pairwise comparisons, confirming that GPA disparities extend beyond extreme cases.

Effect size calculations using Cohen's d ($> 1.5$ in all high- vs. low-achieving cluster comparisons) further reinforced these findings, indicating a very large effect. These results provide strong statistical evidence that digital behavior patterns correlate with academic performance and that GPA differences across clusters are meaningful rather than random variation.

### 3.8 Implications and Limitations

Our findings indicate a clear relationship between students' digital behavior and academic performance within the analyzed dataset. At the same time, since GPA was introduced only at the final stage, the proposed framework is not tied to this outcome alone and could, in principle, be adapted to other educational indicators. This flexibility makes the approach potentially relevant for broader educational analytics, where different target variables may capture complementary aspects of student development.

This study is observational, identifying correlations rather than causation. It remains unclear whether consuming educational content improves academic outcomes or whether high-achieving students are simply more likely to engage with it. A longitudinal study could help clarify this relationship. Another limitation is potential sampling bias, as the dataset includes only publicly available profiles provided voluntarily by students, which may not fully represent the broader student population. In addition, the data comes from a single university, which limits external generalizability.

A further limitation is the absence of a dedicated confound-controlled analysis. Part of the observed variation may therefore reflect differences unrelated to digital behavior alone. Additionally, GPA is not the sole measure of success, and students may have different educational goals and priorities. Future research should incorporate richer educational targets, explicit control for confounding factors, and alternative behavioral metrics such as discussion engagement or interaction patterns for a more comprehensive analysis.

## 4 Conclusion

This study provides empirical evidence that students' digital behavior, as reflected in their social media activity, is associated with academic performance. Using a multimodal clustering approach that integrates textual and visual embeddings, we identified 52 student clusters with distinct academic outcomes. High-achieving students predominantly belong to clusters focused on foreign language learning, science-related discussions, and active university engagement, frequently interacting with educational resources and academic communities. In contrast, low-performing students engage more with informal humor, gaming, and lifestyle-oriented content, showing little academic-related activity.

Statistical analyses confirmed substantial GPA differences across clusters, with ANOVA results indicating highly significant variation ($p < 10^{-100}$). Post-hoc comparisons further demonstrated that students in high-achieving clusters consistently outperform those in low-achieving clusters, with large effect sizes (Cohen's d $> 1.5$). Notably, text-based engagement patterns were more informative for distinguishing academic outcomes than image-based clustering, highlighting the importance of semantic context in digital interactions.

Overall, the proposed framework demonstrates the potential of multimodal educational analytics for identifying structured patterns in students' online behavior. At the same time, the results should be interpreted within the limits of an observational study based on data from a single university. Future research should extend this analysis to richer educational indicators, longitudinal settings, and broader student populations.

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
