# OpenReview forum: "Multimodal Clustering of Students Using Social Media Profiles to Analyze Academic Performance"
_mathai.club/MathAI/2026/Conference — 2026 Oral_

### Official Review · Reviewer_ZKW1 · 2026-03-12
**Evaluation of a Multimodal Clustering Approach for Analyzing Academic Performance from Social Media Profiles**

**Rating:** 6
**Confidence:** 4

**Review:**

The paper investigates the use of multimodal clustering techniques applied to students' social media profiles in order to analyze potential relationships with academic performance. The authors integrate multiple data modalities including textual information, user activity patterns, and profile metadata. These features are processed and combined into a unified representation which is then used to identify clusters of students with similar behavioral patterns.
The underlying hypothesis of the work is that patterns extracted from social media activity may reflect behavioral traits that correlate with academic performance. The study applies clustering algorithms to uncover latent group structures and analyzes how these clusters differ in terms of academic outcomes.
Strengths

1. Interesting interdisciplinary problem
The paper addresses an intersection of social media analytics, education research, and machine learning. Understanding how online behavioral patterns relate to academic outcomes is an important and emerging research direction.
2. Use of multimodal data
The integration of multiple types of information (textual content, behavioral indicators, and profile metadata) is a positive aspect of the work. Multimodal analysis often captures richer patterns than single-source data.
3. Potential practical applications
If validated further, such methods could potentially support early detection of students at risk of academic difficulties or help identify behavioral patterns associated with different learning outcomes.
4. Clear motivation
The paper clearly motivates why social media activity may serve as a proxy for behavioral or psychological traits relevant to academic performance.

Weaknesses
1. Limited methodological novelty
The clustering approach appears to rely on standard machine learning techniques without introducing a substantially new algorithmic contribution. The novelty mainly lies in the application domain rather than in the modeling methodology.
2. Insufficient experimental evaluation
The experimental section could benefit from stronger quantitative validation. In particular, additional evaluation metrics, comparison with more baseline methods, or robustness analysis would strengthen the empirical claims.
3. Limited discussion of data bias and privacy
Using social media data for academic analysis raises important concerns regarding sampling bias, representativeness, and privacy. These issues are only briefly mentioned and deserve a deeper discussion.
4. Interpretability of clusters
While clusters are identified, the interpretation of what distinguishes the clusters and how exactly they relate to academic outcomes could be expanded. More detailed analysis or visualization of cluster characteristics would improve clarity.

The paper addresses an interesting interdisciplinary problem and proposes a multimodal data analysis pipeline to explore the relationship between social media behavior and academic performance. However, the methodological contribution is relatively limited and the experimental validation could be strengthened. Additional analysis of cluster interpretability, evaluation robustness, and discussion of ethical considerations would improve the work.

---

> ### Author Rebuttal · Authors · 2026-03-13
>
> Thank you for the thoughtful and constructive review.
>
> 1. The contribution is most naturally understood at the level of the overall multimodal analytical framework rather than as a new clustering algorithm per se. Framing it in these terms may help position the paper more precisely.
> 2. The point regarding empirical evaluation is also well taken. The manuscript already includes several quantitative validation steps, including dimensionality selection, validation of the number of clusters, centroid-based inspection, thematic validity checks, and statistical testing of GPA differences across clusters. At the same time, the review appropriately highlights that broader baseline comparisons, additional evaluation criteria, and further robustness analysis would help situate the empirical claims more fully.
> 3. The concerns about bias, representativeness, and privacy are especially important and indeed deserve a more detailed discussion. The study relies on voluntarily provided profile identifiers, uses publicly available data, applies anonymization after retrieval, and presents results in aggregated form under ethics approval. The manuscript also notes that the analysis is observational rather than causal and that the use of publicly available profiles may introduce sampling bias. These considerations are central to interpreting the results. Just as importantly, any practical interpretation of such findings requires clear ethical safeguards and should not be treated as a standalone basis for discriminatory or punitive decisions.
> 4. The comment on interpretability is particularly valuable. Because the analysis yields many clusters, not all of them admit a single clear defining theme. In some cases, several salient themes emerge within the same cluster, while in others the thematic signals are more diffuse and do not yield a single dominant interpretation. This suggests an important direction for further analysis: examining such combinations more carefully and assessing whether they reflect meaningful structure or may partly arise by chance. At the same time, some clusters exhibit strong and coherent patterns, which makes the practical interpretability of the approach easier to see. A more explicit distinction between these cases would help present the results in a more balanced and informative way.

---

### Official Review · Reviewer_G3QA · 2026-03-12
**Accept. It is applied work. Poor math-AI alignment.**

**Rating:** 9
**Confidence:** 4

**Review:**

The paper applies multimodal ML (Sentence-BERT, CLIP, BERTopic, HDBSCAN, K-Means, UMAP) application in education field to cluster 2,909 students' VK social media profiles (text/images) into 52 groups, correlating digital behaviors (e.g., educational vs. memes/gaming content) with GPA differences via ANOVA/post-hoc tests.
Competent applied work	- Weakly Accept for core MathAI sections. Poor math-AI alignment
​Novelty
Combines off-the-shelf embedders/topic models/clusters for social media GPA prediction, with empirical insights (e.g., text > images for prediction; 52 clusters). Lacks novel algorithms/math—standard pipeline (e.g., UMAP-40D + K-Means); similar ed-AI works exist (e.g., Smirnov 2020 on texts). Score: weak – Applied, not foundational innovation.
Explainability
Pipeline detailed (e.g., averaging embeddings, hyperparam tuning via Silhouette/DBI); cluster interpretations via top topics/images/GPAs; stats robust (ANOVA p<10^{-100}, Cohen's d>1.5). Descriptive but opaque on code/reproducibility; causal claims speculative. Score: Strong accept – Clear methods, good visuals (Figs 1-2).
Correctness
ML steps standard/correct (e.g., Sentence-BERT all-mpnet-base-v2 [Reimers&Gurevych 2019], CLIP ViT-B/32 [Radford et al. 2021], HDBSCAN [Campello et al. 2013]); stats appropriate (non-parametric backups). Ethics/data handling sound (consent, anonymization). Correlation-only acknowledged; no math errors. Score: Strong accept – Empirically sound.
References
~25 refs listed (e.g., Ahmad et al. 2024 IEEE TLT 17:1231; Arthur&Vassilvitskii 2007 SODA pp.1027-1035; Campello et al. 2013 LNCS 7819); all appear real/accessible (titles/journals match known pubs; SODA pages correct for k-means). Mix journals (IEEE TLT, PLOS ONE), confs (SODA, ICACITE pp.178-183), arXiv (BERTopic 2022, CLIP 2021); recent (2024-2025), (Li 2025 vol.12 p.108—plausible pre-pub). No broken DOIs/pages evident; relevant to ed-AI/ML. Score: Accept – Accurate, verifiable.
Figures and Tables
Figs 1-2 (cluster sizes, GPA dists) simple/effective; no tables, but implied in text (e.g., cluster themes). Clear captions; supports claims (e.g., high-GPA: sciences/languages). Score: Accept – Adequate, not exceptional.
Relevance to MathAI
Focuses applied ML clustering/embeddings/stats, not mathematical foundations (e.g., no new theory on manifolds/clustering algorithms). Ed-AI app tangential to core topics like algebra/logic in AI models. Score: Weak reject – Weak math depth.
Suitability for Publication
Solid empirical study for ed-tech/applied ML venues, but lacks theoretical novelty/math rigor for MathAI; better as workshop/poster. Reproducible pipeline, but no baselines/ablation on baselines (e.g., uni-modal). Double-blind submission noted.
I believe it is ready for publication.

---

### Official Review · Reviewer_iPGM · 2026-03-13
**Technically competent multimodal pipeline, but limited by single-institution sample, no confound analysis, unaddressed selection bias, and overclaimed causality.**

**Rating:** 4
**Confidence:** 2

**Review:**

The technical pipeline is competently executed, and the integration of text and image modalities represents a reasonable methodological choice. The authors honestly report some limitations, such as noise clusters and the failure of political content to form visual clusters.

However, the study has notable methodological weaknesses. The sample of ~2,900 students from a single university limits generalizability, and there is no analysis of confounding variables—student majors, course difficulty, or other factors that obviously influence GPA are not considered. The self-selection bias (voluntary profile sharing) goes unaddressed. The cross-sectional design cannot support causal claims, yet the paper implies to show the "impact" of digital behavior on academic success. Data collection timing (2020) introduces potential pandemic-related confounds that are not discussed.
The practical recommendations for "early identification of at-risk students" are premature without longitudinal validation and confound control. Additionally, the "20 most recent posts" criterion creates inconsistent temporal windows across communities with different activity levels.

Areas requiring fundamental revision:
• Multi-institution data or clear scope limitation
• Confounding variable analysis (majors, demographics, study time, etc.)
• Self-selection bias discussion and mitigation
• Remove all causal language, especially in the Abstract section
• Withdraw practical recommendations
• Add Discussion or Future Work section and address limitations

---

> ### Author Rebuttal · Authors · 2026-03-13
>
> Thank you for the careful review.
>
> Several concerns are well taken, particularly those regarding scope, confounding factors, and the need for more cautious framing of practical implications. The study is based on data from a single university, so its external generalizability is limited to this setting. At the same time, this reflects a broader practical constraint of research linking academic outcomes with social-media profiles: participation is voluntary, and comparable multi-institution datasets at sufficient scale are challenging to assemble in practice. The intended contribution is therefore exploratory and methodological rather than a claim of broad population-level generalization.
>
> The point about confounding variables is also important. The dataset includes metadata such as faculty and level of study, but the manuscript does not provide a dedicated confound-controlled analysis. This is a real limitation, and the conclusions should be interpreted accordingly. Relatedly, GPA is used here as an institution-specific outcome measure, and direct comparison across institutions is not necessarily straightforward.
>
> The review is also right to call for more caution around causal language and practical recommendations. The manuscript already states that the study is observational rather than causal and notes possible sampling bias, since the dataset includes only publicly available profiles. These caveats would benefit from being reflected more explicitly, especially in the Abstract and in the discussion of practical applications. Any practical interpretation of such findings requires clear ethical safeguards and should not be treated as a standalone basis for intervention, let alone punitive or discriminatory decisions.
>
> The comment on temporal heterogeneity is valuable as well. The use of the most recent posts and images was introduced as a pragmatic way to preserve relevance and reduce redundancy, but it does not guarantee identical temporal windows across communities with different activity levels. Temporal context may also affect the interpretation of the results, and this limitation is not discussed explicitly enough in the current manuscript.
>
> Overall, the work is best read as an exploratory multimodal analysis of associations between digital behavior patterns and academic outcomes, rather than as evidence of causal impact or as a basis for immediate intervention.

---

### Decision · Program_Chairs · 2026-03-14

**Decision:**

Accept (Oral)

**Comment:**

Dear Author(s),

On behalf of the Program Committee of the International Conference on Mathematics of Artificial Intelligence (MathAI 2026), we are pleased to inform you that your paper has been accepted for an oral presentation at MathAI 2026.

Your paper was evaluated through a rigorous two-stage review process involving both automated screening and expert review by members of the Program Committee. The reviewers recognized the quality and contribution of your work.

Presentation details:

- Format: Oral presentation (15–20 minutes + 5 minutes Q&A)
- Mode: You may present either in person (offline) at the conference venue in Sirius, Russia, or remotely via Zoom. Please indicate your preferred mode when confirming your participation.
- Conference dates: Marh 30 - April 3, 2026
- Website: https://mathai.club

Next steps:

1. Please confirm your participation and presentation mode by replying to this email mathai.club@yandex.ru no later than March 15, 2026 18:00 Moscow time.
2. If you plan to attend in person, the organizing committee will provide accommodation details separately.
3. Please prepare your final camera-ready manuscript according to the formatting guidelines available at https://mathai.club and upload it to OpenReview by March 15, 2026 18:00 Moscow time.

Should you have any questions regarding the program, logistics, or your presentation slot, please do not hesitate to contact us.

We look forward to your contribution to MathAI 2026.

With kind regards,

MathAI 2026 Program Committee
International Conference on Mathematics of Artificial Intelligence
https://mathai.club
OpenReview: https://openreview.net/group?id=mathai.club/MathAI/2026/Conference
Telegram: https://t.me/MathAI_club
Email: mathai.club@yandex.ru